# Risk Factors for the Mental Health of Adolescents from the Parental Perspective: Photo-Voice in Rural Communities of Ecuador

**DOI:** 10.3390/ijerph20032205

**Published:** 2023-01-26

**Authors:** Esteban Baus, Majo Carrasco-Tenezaca, Molly Frey, Venus Medina-Maldonado

**Affiliations:** 1Centro de Investigación para la Salud en América Latina (CISeAL), Medicine Faculty, Pontifical Catholic University of Ecuador, Nayón 170530, Ecuador; 2Centro de Investigación para la Salud en América Latina (CISeAL), Pontifical Catholic University of Ecuador, Nayón 170530, Ecuador; 3Ohio University Heritage College of Osteopathic Medicine (OUHCOM), Dublin, OH 43016, USA; 4Centro de Investigación para la Salud en América Latina (CISeAL), Nursing Faculty, Pontifical Catholic University of Ecuador, Nayón 170530, Ecuador; 5Research Group of Gender Violence Prevention (E-previo), Nursing Faculty, Pontifical Catholic University of Ecuador, Quito 170143, Ecuador

**Keywords:** mental health, adolescents, risk factors, photo-voice, community participation

## Abstract

Mental health in adolescence is a very important topic worldwide, especially in rural areas. The implementation of Participatory Action Research (PAR) through the photo-voice method was a way to encourage adults to recognize problematic situations (at personal, family or community levels) that threaten the well-being of adolescents, but that in everyday life may go unnoticed by parents and caregivers. Our study aimed to identify risk factors for mental health in adolescents living in rural communities of Ecuador from the parental perspective through photographs and focus group discussions. The study sought to raise awareness of this issue at the family and community levels. The photo-voice method was conducted with the participation of 29 parents. The photographs and the collaborative construction of meanings allowed parents to have a better understanding about the importance of mental health and its benefit for adolescents. The principal risk factors mentioned by parents were stress, sleep deprivation, tiredness, poverty, difficulties in continuing education and alcohol consumption. In conclusion, we point out the importance of this intervention to explore the knowledge and understanding of the topic by parents as well as to communicate information that demystifies false beliefs around mental health.

## 1. Introduction

Currently, the issue of mental health in adolescents is of great importance; it is especially a challenge for those parents who live in rural areas, since some of these community’s face limitations in terms of knowledge of the subject and accessibility to health services. Different international studies show that more than 80% of adolescents live in low- and middle-income countries [1]. From an epidemiological point of view, in Latin America and the Caribbean almost 16 million adolescents aged 10 to 19 years old live with a mental disorder, of which approximately 50% are due to anxiety and depression. In terms of mortality, suicide is the third leading cause of death among adolescents aged 15–19 in the region [2].

In Ecuador, rural contexts are a challenge in terms of understanding what adolescents want, what motivates or worries them and how the city influences their social dynamics along with their decisions. Approaching the perception of their reality through the parents’ perspectives will allow us to identify possible risk factors for mental health in this segment of the population. Furthermore, the activity was an important contribution to the national health system. It presents deep limitations that translate into barriers to access mental health services such as shortages of mental health professionals, the scarce participation of the family and the community in activities to promote mental health and in the prevention of mental health problems and the care and follow-up of users with mental issues [3] (Baena, 2018).

From the theoretical point of view, risk and protective factors are widely used in public health for both the assessment and design of adolescent health interventions [4,5,6,7]. One of the most widely used approaches for the determination of these factors is the ecological systems theory of Bronfenbrenner (1979) where an individual is seen as a system that is composed of other subsystems and the dynamics between the biological, psychological and social within a context are recognized [8]. Understanding the complexity of interactions between risk and disease can be difficult but using life course risk modeling for mental health can help to further understand the trajectories from risk to disease. Notably, key risk factors from widespread diagnostic categories include low socioeconomic status, familial psychopathology, stressful life events, low IQ, family dysfunction and cannabis use. These factors do not dive into the exact mechanism of the transition from risk exposure and later mental health adversities unlike life course risk modeling. Life course risk modeling describes two periods, critical periods, and accumulated risk periods that can have a direct correlation to later adverse mental health outcomes. The critical period model assumes that during certain stages of human development, such as adolescence, external factors may have critical effects that can lead to later disease. The accumulation of the risk period suggests that over time different experiences and exposures can lead to an increased risk of disease. Life course risk models recognize the importance of timing and the cumulative nature of risk for disease ultimately advancing our understanding between risk and disease outcome [9].

A recent systematic review based on intervention studies recommended the preparation of parents to play an active role in promoting the mental well-being of children and youths [10]. It emphasized the importance of mental health literacy, self-care promotion, positive reinforcement, the reduction of harsh discipline in parenting, the consistency of parenting, as well as social support. It is also important to incorporate studies on adults’ opinions and feelings about these issues and the implementation of education programs to improve their skills to support their children [11].

The implementation of Participatory Action Research (PAR) through photo-voice in this project is a way to encourage adults to recognize problematic situations (at personal, family or community levels) that threaten the well-being of adolescents, but that in everyday life may go unnoticed by parents and caregivers.

This study aimed to explore the risk factors for the mental health of adolescents from a parental perspective using focus group discussions and photo-voice. Our research question was, “What can be achieved in the understanding and transformation of knowledge related to risk factors for adolescents’ mental health by their parents based on Photo-voice?”

## 2. Materials and Methods

This qualitative study was based on Participatory Action Research (PAR) with a methodological focus on the exploration of concrete social needs to guide action. Particularly, the information on the concern of the members of the community was examined. The action was developed to strengthen the capacities of adults in recognizing the factors that can be a threat to the mental health of adolescents in rural communities [12].

The photo-voice method was theoretically developed from three sources: the literature or education for critical consciousness, feminist theory and documentary photography [13]. In this study, a dialogic orientation was privileged for the transformation of knowledge [14] and empowerment of participants involving leading individuals through a set of focus group questions and the assistance of photos [15]. In terms of mental health education, photo-voice in this study was used as an intervention; the group of participants reflected on their own concerns and their role as parents in fostering meaningful connections and the importance of talking about problems that could block the promotion of mental well-being in adolescents of three rural communities in Ecuador.

For participants’ selection, researchers implemented an intentional non-probabilistic sampling that, according to the purpose of the research, responded to a necessity felt by the parents of the communities. The method of approach was face to face with field visits, which corresponds to the form of interaction between the members of the locality. The sample size was 29 individuals in 5 focus groups. At all times, the principle of voluntariness and the right to refuse participation in any phase of the investigation was maintained.

Inclusion criteria for participation were adults older than 18 years of age; fathers, mothers or guardians of adolescents; members of rural communities or adults who served as natural, cultural leaders and legal or political representatives. Participants declared their acceptance by signing informed consent forms. All individuals participated in orientation on safe and ethical photography.

The data were collected in three communities in Loja province, southern Ecuador: Guara, Chaquizhca and Bellamaría. They have been involved in a program that carries out activities for the control and prevention of Chagas disease for more than 15 years [16]. Each community has a single-teacher school that children attend until they start their high school education in the nearest town, Cariamanga. At this point, they move to Cariamanga and return to their communities over the weekends or they travel to the town each morning and return in the afternoon with a commuting time of at least an hour. There are some health promoters in each community, but the closest health center is located in Cariamanga [16,17]. Due to the travel time, access to health and education services is a challenge for the inhabitants of the three communities.

Most of the focus group discussions took place in the classrooms of the three schools. Due to the characteristics of the community, some focus groups were held in the houses of the participants during the first phase of the study. The implementation of the photo-voice activity took place in the surroundings of the community, where the lives of the subjects under study normally take place.

The analysis of the risk factors in the personal, family and community dimension led to a discussion and the abstraction of concepts through photo-voice. The discussion began with the focus group whose purpose was to explore the understanding of the term mental health. Once their understanding was verified, the central theme was discussed based on the research question: “What are the risk factors for the mental health of adolescents from the perspective of parents or members of the community?” Each intervention was argued by the focus group participants, validated and enriched by the rest of the participants.

The second phase of the study consisted in applying the photo-voice taking into consideration the risk factors identified and validated in focus groups. Then, the participants were asked how to capture the aspects discussed in photographs without this representation including people, intimate situations or situations that violated the privacy of individuals. During the discussion, the members of the community represented the categories analyzed by applying the SHOWED technique to determine the message of the images that were photographed [18].

Data collection was carried out for a period of two weeks in the month of June 2022. Participant observation, a transversal axis of the research, led to the immersion of the research team in activities to document the behavior of the participants over the course of these days, contrast narratives with practice, reduce the incidence of “reactivity” and to develop culturally relevant questions [19]. Similarly, the research team provided advice to the participants on the subject of mental health when required, made referrals of people with mental health problems to the health services in the nearest city and carried out the subsequent follow-up.

During the data collection, some foreign students and teachers belonging to the Tropical Disease Research and Service-Learning programs (TDR-2022) from Ohio University and Ohio University Heritage College of Osteopathic Medicine were present. They observed the exchange process with the members of the community as a complementary activity to their academic training.

Regarding demographic characteristics of parents, the age of participants ranged from 26 to 47 years old, with more than half participants being female. Regarding occupation, most of the men reported to be farmers and women homemakers.

The research team considered themes and categories *a priori* using a deductive approach derived from the empirical framework and the qualitative contents analysis [8,20]. In our study, the coding frame assigned the same level of specificity to each category (See Table 1).

Subcategories emerged in an inductive way and they derived from Deductive Category Assignments.

During fieldwork, a preliminary sounding was carried out on the research question and risk factors for the mental health of adolescents. Researchers designed the question guide used in the focus groups: How would you describe the personal aspects that affect the mental health of adolescents? What would be your opinions in relation to the family situation that affects adolescents? What would be your opinions in relation to the community situation that affects adolescents?

The verbatims were recorded and stored in voice files using digital recording equipment, then transcribed with NVIVO software. The recordings were listened to again and the transcripts edited by the researchers. This facilitated the process of reading, rereading and analyzing the documented interactions. Focus group times ranged from 30–60 min with an additional 15 min on average during the photo shoot process. The saturation of the information was obtained in the focus groups once the topic of discussion had ended and as no new perspectives of the analyzed situations were found.

We used qualitative content analysis [20] with the application of deductive categories that determined exactly under what circumstances a passage of text could be coded with a category and the development of inductive categories derived from the reality experienced by participants that complemented the meaning of the phenomenon under study. For the treatment and systematization of the information, ATLAS ti version 7.2 was used with a license acquired by the Pontifical Catholic University of Ecuador.

The protocol was approved at the Ethics and Research Committee of the Pontifical Catholic University of Ecuador. All necessary measures were considered to protect confidentiality, present the information anonymously, as well as the use of pictures and meanings. Similarly, the principle of respect for the autonomy of the participants, voluntariness and the right to interrupt the study when the subjects so wished prevailed.

## 3. Results

The subjects were identified using the letter “S” for SPEAKER and a code number to differentiate each participant, followed by the acronym “FG” corresponding to focus group and a number to identify each group. Regarding the dynamics of participation, the first two groups were more reserved when expressing their ideas compared to the following three.

The one topic discussed in focus groups was: risk factors in adolescents from rural communities. Then, researchers developed subordinate categories and subcategories based on deductive and inductive analysis. We obtained an absolute count of 3 deductive category assignments and 22 inductive subcategories from 5 documents. However, we presented a resume of subcategories based on the more common themes in participants’ responses (grounded) and the linkages to other subcategories (theoretical density). It was obtained from the Atlas ti code manager. The numbers in curly brackets indicated {x- the quotes linked to each subcategory and the second number x} the number of times in which this subcategory was linked to other.

The accompaniment of researchers previously known by members of the community had a positive impact on the exchange, the development of ideas and the identification of meanings to determine the message of the images that were photographed.

### 3.1. Risk Factors in Adolescents from Rural Communities of Ecuador

This theme shows the results from the parental view that can affect or undermine the healthy development of adolescents.

#### 3.1.1. Personal Risk Factors

In this deductive category, the parents expressed their ideas about those circumstances that are inherent to the individual dimension and affect the adolescent’s mental health.

One of the most relevant subcategories in the life of adolescents from the parental perspective was the constant load of *stress* {11-2} due to external and internal demands related to personal characteristics or educational institutions.

“*Here they did not experience so much stress, since they have to go to the city they have more*”.(S5-FG3)

“*They stress because they are given lots of homework and they need to do the homework on time. If they don’t have help then they get stressed and feel sad because they think they can’t do something or that they don’t have the capacity to do it*”.(S6-FG3)

The next subcategory was *sleep deprivation* {12-3}; most recognized the decrease in quality and hours of sleep. Here are some excerpts from the discussion:

“*They are tired because I have also gone with my children, they get up at four in the morning to have breakfast and if we go past five minutes it is already late. They come home at four in the afternoon, they come tired, wanting to sleep for a whole day, right after one asks them to come help out with something in the house and they no longer want to, they arrive, eat at that time, do homework and they stay up late, they don’t rest well*”.(S1-FG1)

“*My daughter is going to study in the city, she has to get up early at 4 in the morning because at five she is already leaving*”.(S8-FG4)

Both *stress* and *sleep deprivation* subcategories were related to the disadvantage of not being able to continue with secondary education in their communities and having to move to a distant place to be able to do it. The farther the rural community was, the more frequent the experiences of economic limitations and school dropout were.

Other personal risks recognized within the narratives by the parents in the focus groups were grouped into the following subcategories:

The subcategory *behavioral change* {7-0} was interpreted as those signs that make it easier for parents to perceive a difficulty in the life of the adolescents.

“*You can see their behavior because sometimes their mood changes*”.(S4-FG4)

“*But on the other hand, if ever when the son feels bad or something happens one realizes, it is for the reason that we say he is quiet, calm or sad, we ask them, why do you feel sad, what is wrong with you or what do you need*”.(S6-FG4)

“*And of course they are active and suddenly if they go elsewhere they isolate themselves. Then one already realizes it*”.(S2-FG5)

According to what was described, the situations that could trigger the subcategory *sadness* {5-2} mostly were low grades, but it was also understood as a sign to support adolescents.

“*If they have problems at school, they can’t give us good grades at the beginning of the school year or the end, so they get sad, depressed*”.(S3-FG4)

“*We (mothers) know when our children are sad*”.(S5-FG5)

The subcategory *low self-esteem* {4-3} was also identified by parents as a personal risk for the mental health of adolescents in the rural community.

“*Disagreement with their own bodies*”.(S1-FG2)

“*When they have very low self-esteem*”.(S2-FG2)

Finally, the narratives of the parents in several focus groups identified *cognitive distortions* {5-2} as a risk factor that was related to the lack of skills to make the right decisions or make choices that would help adolescents to stay healthy. In this particular risk, the narrated experiences pointed towards alcohol consumption. During discussions, parents were aware about the local situation and how social determinants can affect mental health in adolescents. Parents understood the importance of this problem and why it should be addressed by families and community members and even with local government.

#### 3.1.2. Family Risk Factors

The experiences mentioned by the participants facilitated reflection on parental functions and family dynamics that negatively affect the mental health of adolescents. Some situations expressed by participants were listed: reprimanding without purpose (normalized violence, maltreatment); the lack of care during the upbringing of children (a risk factor because it can lead to lashing out at a child) and the lack of harmony amongst family.

The *inadequate supervision* subcategory {8-1} corresponds to the lack of the protective function that the family should fulfill with its members; the discussion in the focus groups was oriented towards the reflection of the parents not only on the sustenance needs of adolescents, but also towards support, caring and the orientation that should be provided.

“*Lack of care and attention*”.(S3-FG1)

“*It may also be that, for example, one is not taking care of the children and they look for someone other than the father or mother. That can be a bad thing*”.(S5-FG5)

Regarding the subcategory *lack of affection* {6-1}, parents concluded that the absence of expressions of love and the non-acceptance of adolescents are also a risk to their mental health.

“*It may be that they do not want him in the family*”.(S3-FG2)

“*Lack of family affection*”.(S2-FG2)

“*Having differences in treatment between children*”.(S3-FG3)

“*Maybe they think they don’t love them*”.(S2-FG5)

During the narratives, the *intimate partner violence* subcategory {5-0} was identified as a problem in the conjugal dyad. It was also considered a risk factor for the mental health of adolescents; this abusive environment breaks normal family functioning.

“*The problems that happen inside the house*”.(S5-FG3)

“*So, let’s say the dads fight and can’t talk about anything normal*”.(S4-FG3)

“*When there is not a good relationship between husband and wife*”.(S3-FG5)

“*But also fights between couples affect and lead to getting angry with them for no reason, having arguments for no reason*”.(S3-FG5)

#### 3.1.3. Community Risk Factors

Parents reflected on and highlighted the external circumstances that could make adolescents vulnerable to mental problems. This section shows how factors at the community level affected the individual level.

In the narratives of the *norms favorable to alcohol use* subcategory {14-2}, it was observed how culture can encourage risk behaviors. The participants commented that their perceptions about collective behavior facilitated engagement in alcohol consumption from a young age.

“*Of course, here adolescents from the age of 14 practically already drink alcohol*”.(S1-FG2)

“*The truth is that here yes, in all the communities here, from the age of 14 they already use what are drinks, drinks, tobacco, everything*”.(S3-FG2)

“*I sell alcohol, but minors buy a few because they don’t have money to buy anything if the adult provides it and whoever drinks that, and if not, they don’t have money to buy. With the fair ones he goes to the school that they give to his parents. I believe that the parents are not going to give their sons 10 dollars in cash so that he can go drink*”.(S6-FG4)

Observation: Researchers reinforced the responsibility of participants to avoid sharing alcohol with adolescents.

The following testimony can be interpreted as the normalization of the use of legal drugs by adolescents in rural communities.

“*Here the consumption is alcohol or tobacco but not drugs*”.(S6-FG4)

The lack of *preventive education* subcategory {2-0} was mentioned because of the necessity for programming implementation to promote mental health in adolescents. Participants expressed recommendations about psychoeducational prevention in topics such as self-harm, suicide, alcohol and other drug use.

“*Respect the body, do not harm the body. Teach them to respect their bodies*”.(S4-FG1)

“*I say that it is important to have information aids, that is, informative talks to adolescents about how bad it is to consume alcoholic beverages, drugs, all that, talks because there is nothing here and I think practically nowhere*”.(S1-FG2)

The *lack of transport facilities* subcategory {16-1} reflected on how the communities are located about 15 km from the nearest city. The road is dirt, rugged and passable during the summer season. In the winter season, the road becomes impassable due to the mud and few vehicles manage to pass. It receives sporadic maintenance. The inhabitants of these communities travel to the city in open buses, called rancheras, leaving at 5:00 am and returning at 4:00 pm; the cost is about USD 1.00. The absence of public transport on a regular basis is one of the greatest difficulties experienced by adolescents with aspirations to continue high school. This lack of transport facilities at the community level affects adolescents on a personal and family level.

“*(…) Transportation, we pay for that and it does affect us financially*”.(S7-FG4)

“*Teenagers have to make a little more effort to study because of the distance*”.(S3-FG5)

“*So, my nephew already told him: no daddy, I don’t study anymore, why? If you don’t have enough time to work, you don’t have to pay, if you don’t have money to pay the tickets, how am I going to go?*”.(S6-FG3)

The following testimony refers to the difficulties and risks within the rural community itself in moving from one place to another.

“*And for the sector where I live, I don’t live for this sector but for the other side of the creek, because I live there with my brothers and I don’t know who wants to do evil, that is, and for that reason. I feel sad to see my son, because it scares me that something might happen to him along the way*”.(S8-FG3)

The transportation and secondary education needs in these communities contribute to a sense of neglect by the state.

“*We need to have the school closer or maybe have a collective transport*”.(S2-FG1)

“*So far this year, we pray to have secondary education here, we ask for transportation, but not anymore, there is no solution. If we talk to the mayor, let him be the one who supports us with half the cost of transportation, for a few days he supported us to cover these study expenses in the boys’ studies, they gave us half, but not now*”.(S7-FG3)

The *limited access to secondary education* subcategory {8-1} shows the situation in the communities of Guara, Chaquizhca and Bellamaría for adolescents:

“*They get up early to receive the education, now is different as it was, also they need transportation from here to Cariamanga and it is far away*”.(S8-FG3)

“*We want something different and sometimes we believe that everything is possible. We tried to put the high school here, we tried it with the teacher, they looked for documents, the time passed without answer, other families continue to spend much more money and we didn’t find a solution*”.(S5-FG3)

The following expression is an example of adults’ socialized aspirations and motives for supporting their sons’ education.

“*They say it’s better to work, but at the same time it’s a shame because it happens to them like someone who only finished primary school, and no one knows more about it*”.(S8-FG3)

The *lack of economic opportunity* subcategory {17-2} showed the vulnerability of families and community members. This was a distinctive risk in relation to income and economic stress. This aspect affected people from all age groups.

“*When there is concern that there is no way to spend the money, when they see the father or mother suffering. How are they going to get their money and now how do they go to school, they already ask for this and there is none, so they think it is better to be working*”.(S7-FG3)

“*They decide and want to continue studying, but seeing how we are at home, they prefer to work and the adolescents say that it is better to help improve family life. But they don’t do it because they want to. The same to give them time at home to study. They can’t at the same time want to study, but as you can see there isn’t. There is no way to give them*”.(S5-FG3)

“*Well, let’s say the conditions are not very good, (...) because in reality there is a lack for study, a lack for medicine, (…) with money there is everything, with money there is education, everything that is needed. But instead now, with the poor, well, one lacks everything, (…) it lacks for education, for medicine. Maybe even to eat. So if one is missing*”.(S8-FG4)

“*But for example here let’s say people study so sometimes they have something like a ticket for food and on the other hand there are also those who don’t, no, we don’t have, so we can’t give them the study like this*”.(S4-FG5)

### 3.2. Photo-Voice

The reality observed in the focus groups was complex. Participants thought that some situations were better represented with people as models in the photography. However, researchers spoke about restrictions to photographing people. The reflective discussion between participants raised other ideas during the photo-voice phase. Before the activity, participants selected possible images with their meaning to represent risk factors that were affecting the healthy development of adolescents. Then, they took the photographs or directed the shots, and they provided additional meanings around these images through metaphor, storytelling and symbolism (See Figure 1). Through their active participation, this phase was closely linked to creativity, awareness and parental skills learning.

The schemes have been listed as: (a) *Temptation*: Adolescents and youths are invited to consume alcohol by adults in the community. Adults in the community should encourage adolescents to avoid consuming alcohol; (b and c) *Education*: The lack of secondary education in the community is a stress factor for adolescents and their families. The community’s proposal to solve this stressor is to use the already existing school infrastructure to allow for secondary education in addition to elementary education; (d) *Ranchera*: The Ranchera represents the means to attend school. It is considered a risk due to the distance of the 2 h journey from rural communities to the city. In addition, because it affects the family economy, many adolescents stop attending school due to a lack of resources (transportation, family income and poor road conditions).

This was a way to foster community members’ and local authorities’ sensitization. The photographic exhibition was held as a closing activity in the communal house of one of the rural localities and transportation was provided to facilitate the participation of parents from the most remote communities. A representative of the local government was invited to this activity to promote dialogue between both parties. Additionally, the local government representative supported the mobilization of a psychologist specialized in adolescent care to give a workshop on “The role of the family and the community in the prevention of suicide in adolescents”.

Observation methods were useful to researchers during the photographic exposition. We observed that the audience (community members and a representative of the local government) was sensitized with the meanings of the exhibition. This activity was concluded with an educational message provided by researchers. It was oriented to facing risk factors in mental health with knowledge and concern and had positive implications for parents as well as members of the community.

## 4. Discussion

This study aimed to explore the meanings of risk factors for the mental health of adolescents from the parental perspective. The findings raise the question of whether the parents achieved an understanding and transformation of knowledge through the topic discussion using focus groups and photo-voice.

The collaborative construction of meanings during focus group dynamics and photo-voice activities allowed us to identify that parents are aware of risk factors for the mental health of adolescents in rural communities, but other groups of parents understood and learned the importance of this topic based on the answers of their neighbors or contributions of the research team. The approach used by researchers was stimulated through reflective practice. Some benefits of this practice include moving more toward humanistic paradigms, the underpinning of community capability, the gain in knowledge of familial or local needs and community involvement in targeted activities [21,22,23].

At the personal level, the participants in our study described risk factors such as stress [24] and sleep deprivation related to pressures of the education system [25]. This is also because of communitarian difficulties relating to following a secondary school education. This condition can increase the likelihood of developing diseases. A previous study mentioned ”there is now compelling evidence that chronic stress, especially in childhood, sets in motion a chain of events that translate into heightened risk of mental and physical health problems in later life, including cardiovascular disease, autoimmune disorders, depression, post-traumatic stress disorder, and premature mortality” [26]. These sensitive periods are times of development when life experiences can have a greater impact on outcomes than at other periods during the life course [27,28].

From the perspective of parents in our research, several situations were identified as risk factors such as the lack of love in a family, the lack of understanding, the lack of empathy for adolescents, as well as the lack of education/information for parents to help teenagers understand mental health. Mental health disorders in adolescents [10,11,29,30] related to problematic familial styles and dynamics characterized by unbalanced, rigid or uncaring behaviors have been widely studied. For this reason, assessment is very important to understand family difficulties and thus improve family functioning [31].

In terms of community factors, some of the observed changes were linked to the mesosystem level [8], especially the interactions between families, their neighbors and local government. The photo-voice exhibition had a relevance to the socialization of the risk factors in the mental health of adolescents, as well as the approach to challenges that impact their full development in the macrosystem, referred to as cultural and ideological aspects [8]. Parents during focus groups expressed their concerns that teenagers were introduced to alcohol consumption at 14 years of age. The photographic exhibition was an opportunity to discuss beliefs and attitudes around alcohol consumption of adults in the community, and their negative implications for adolescents’ mental health [32].

In this sense, the mental health approach cannot be seen solely as the provision of services because it hinders the sense of intersectionality. Intervention in determinants of mental health requires the participation of state and private institutions, civil society organizations, communities and families. In essence, it is about promoting social activities that favor support for adolescents and fostering practices so that members of the community participate effectively, actively and responsibly in decisions about health care and its implementation [33].

The strategic relationship between communities and representatives of the local government requires more accompaniment from external actors because most of the community risk factors in mental health will need community empowerment, participation, advocacy as well as knowledge. The building of capability in the community requires orientation and strengthening through volunteer work and contributions from IVS. Indirect supporting evidence of these results is a recent study about how the rural drinking water sector came into being and continues to operate via the synergies of myriad state and nonstate, domestic and international actors and agencies [34]. While the support provided by local government was the preventative program of suicidal prevention, there are efforts that require direct financial support from government, i.e., health assistance, psychological therapy, access to better transportation in the community, support to enhance the school infrastructure and/or financial support for families in the rural communities for the transportation of adolescents to secondary school.

A limitation of this study was related to the knowledge transformation processes from parents of adolescents using photo-voice. The researchers applied *participant observation* and they noted in subjects certain facts such as problem analysis at the personal, family and community level that helped participants to understand the term *“risk factors”* in mental health. Furthermore, participants also showed an understanding and awareness about the importance of the problem, the ability to search for possible solutions through the photo-voice activity as well as the willingness to discuss their needs with the local government representatives. This was based on reaching a consensus in the triangulation process between researchers. However, for future investigations a documentation of the experience is recommended during field work that considers predetermined categories about the process of collective knowledge change. We included in the Appendix A an approximation for this process based on a theoretical framework (Tamara Kavytska, 2022) and a qualitative contents analysis method [20,35].

On other hand, knowledge transformation and the most important aspect, *“practice transformation”*, requires a long follow-up as an opportunity to offer feedback while informing parental skills in sensitive topics such as sexual abuse, self-directed violence, teen dating violence and the dysfunctional family. These themes were displaced by the community risk factors detected during focus groups that caused stress and difficulties in the healthy development of adolescents in rural communities, especially educational access, transportation and alcohol consumption.

## 5. Conclusions

The approach was useful and effective in dealing with issues that were not widely approached previously because mental health in adolescents can be considered taboo for parents in rural communities. This intervention was an opportunity to explore knowledge and understanding of the topic by parents as well as to communicate information that demystified false beliefs around mental health. The longevity of the follow up and the implementation of other activities related to the role of the family and community members in the mental health promotion of adolescents is important. A follow-up should be accompanied by complementary work with adolescents. A psychoeducational program that provides information about the risks that adolescents face and the skills to overcome them would be beneficial. The building of community capability requires long-term orientation and strengthening as well as interventions that remove or reduce those factors that cause preventable negative mental health issues.

## Figures and Tables

**Figure 1 ijerph-20-02205-f001:**
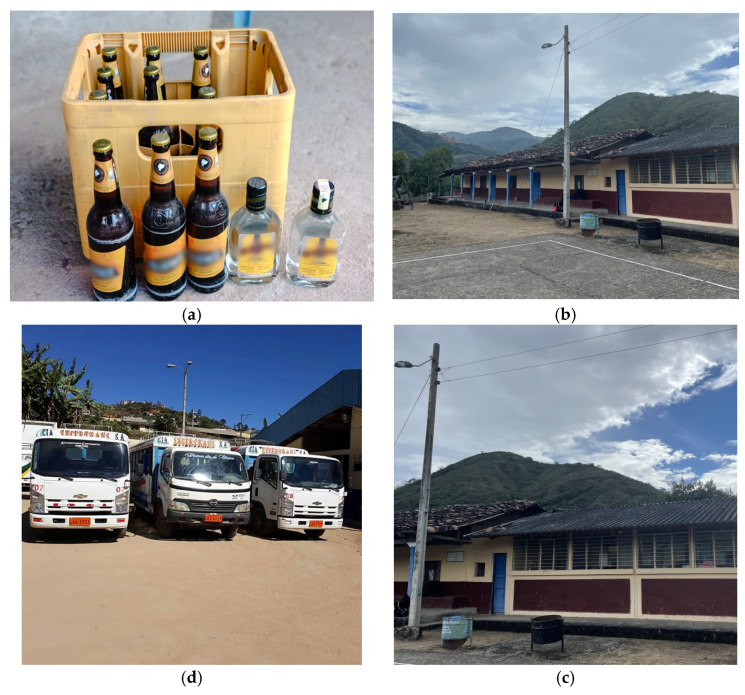
Pictures and meanings obtained during the photo-voice phase.

**Table 1 ijerph-20-02205-t001:** Coding frame.

Deductive Category Assignments	Definition	Anchor Examples
Personal risk factor	Parents can identify those individuals’ behaviors that make adolescents more likely to develop a mental health issue.	“He or she looks stressed”; “He or she consumes alcohol”; “He or she looks depressive”
Family risk factor	Parents can identify problems in family functioning relating to cohesion, routines or communication that make adolescents more likely to develop a mental health issue.	“There is abuse or maltreatment in the family”; “There is inadequate supervision”
Community risk factor	Parents can identify norms favorable to alcohol or substance use and a lack of economic opportunity that make adolescents more likely to develop a mental health issue.	“Adolescents drink alcohol with adults”; “We have problems with our economy”

## Data Availability

The data are not publicly available due to the consent provided by participants on the use of confidential data.

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
