# Peer review of "Risk Factors for the Mental Health of Adolescents from the Parental Perspective: Photo-Voice in Rural Communities of Ecuador"

_ijerph, 2023, doi:10.3390/ijerph20032205_

Round 1
Reviewer 1 Report
Mental health in adolescence is a very important topic worldwide, especially in rural areas. This research aimed to identify risk factors for the mental health in adolescents and young people living in rural communities of Ecuador from the parental perspective through photographs and focus group discussions.
The authors proposed a study seeking to raise awareness of this issue at the family and community level.
Their qualitative study was conducted through Photovoice in which 29 individuals participated. The techniques used were Focus Group Discussions and Participant Observation.
The photographs and the collaborative construction of meanings allowed parents to have a better understanding about mental health and its benefit for adolescents and young people’s stress, sleep deprivation, tiredness, poverty, difficulties in continuing education, and alcohol consumption.
The study is interesting.
I also think that the focus groups are robust means in this field.
I have some minor comments.
1. Please improve the abstract. It is too short. Furthermore, it must better valorize the study and summarize the sections.
2. “What can be achieved in the understanding and transformation of knowledge related to risk factors for adolescents and young peoples’ mental health by their parents?” It is ok to add a research key question. I suggest to enclose also the used tool. For example “ their parents, using and approach based on…..?”
3. Results are interesting. Perhaps the way of presenting them could be improved using some editorial tool, such as boxes or tables.
4. Where is figure 1?
5. Please describe figure 2 in details in the body of the manuscript.
6. The limitations are clearly described “A limitation of this study was related to the knowledge transformation processes from parents of adolescents and young people using the photovoice technique”
7. There is something as suggestion for the future “However, for future investigations it is recommended during the field work a documentation of the experience that considers predetermined categories about the process of the collective knowledge change” I suggest to insert something (structured) on how you suggest to scholars to continue this interesting approach
Author Response
Response to Reviewer 1 Comments
Point 1: Please improve the abstract. It is too short. Furthermore, it must better valorize the study and summarize the sections.
Response 1: Thank you very much for the recommendatios. Done. (in red) Line 19-33
Point 2: “What can be achieved in the understanding and transformation of knowledge related to risk factors for adolescents and young peoples’ mental health by their parents?” It is ok to add a research key question. I suggest to enclose also the used tool. For example “ their parents, using and approach based on…..?”.
Response 2: “¿What can be achieved in the understanding and transformation of knowledge related to risk factors for adolescents and young peoples’ mental health by their parents based on Photo-voice?”(in red) Line 90-91.
Point 3: Results are interesting. Perhaps the way of presenting them could be improved using some editorial tool, such as boxes or tables.
Response 3: We included improvements without editorial tool. We hope the information will be better understand.
Point 4: Where is figure 1?.
Response 4: The mistake was corrected. (in red) Line 395.
Point 5: Please describe figure 2 in details in the body of the manuscript.
Response 5: Done. (in red) Line 396-405.
Point 6: The limitations are clearly described “A limitation of this study was related to the knowledge transformation processes from parents of adolescents and young people using the photovoice technique”.
Response 6: Thank you very much.
Point 7: There is something as suggestion for the future “However, for future investigations it is recommended during the field work a documentation of the experience that considers predetermined categories about the process of the collective knowledge change” I suggest to insert something (structured) on how you suggest to scholars to continue this interesting approach.
Response 7: Done. (in red) Appendix.

Reviewer 2 Report
The abstract refers to photovoice and then 2 techniques - Focus Group Discussions and Participant Observation. Photovoice is a technique plus it is unclear how it is used with the other 2 techniques. From the reading of the method they seem to be separate parts yet they could/should be intertwined more explicitly to elicit deeper understandings from the participants. Also it is unclear how the photovoice activity is related to the results about mental health.
The authors use the phrase "adolescents and young people" however this is misleading as adolescents ARE young people. Recommendation is the use of the term young people as it is more inclusive of a wider age range given that the participants (i.e. parents) could be gaining insights that would be useful for their children mat any age.
The presentation of the data is challenging as the reader is expected to make some assumptions about the value/importance of the quotes used.
There is need for a little more description of each sub-section in the Results as they are not necessarily self evident as related to mental health or wellbeing. Initially both a positive and negative perspective of mental health is offered but it is unclear as to how the authors want to frame the concept. From this it would then be possible to present the rest of the data as either supports for positive mental health versus those circumstances that undermine.
What is a Familiar Risk Factor (see 3.2.2.)? Familiar to whom? or is it menat to be Familial?
In the conclusion the authors claim "A psychoeducational program that provides information about the relevant skills for the risk that adolescents and young people face would be beneficial." While at one level this could be read as a way to build young people's resilience. However another intervention could be about removing or reducing those factors that cause preventable negative mental health issues.
Author Response
Response to Reviewer 2 Comments
English language and style / Moderate English changes required
Response 1: Done.
Point 1: The abstract refers to photovoice and then 2 techniques - Focus Group Discussions and Participant Observation. Photovoice is a technique plus it is unclear how it is used with the other 2 techniques. From the reading of the method they seem to be separate parts yet they could/should be intertwined more explicitly to elicit deeper understandings from the participants. Also it is unclear how the photovoice activity is related to the results about mental health.
Response 1: Thank you very much for the observation. We removed from abstract “photovoice and then 2 techniques - Focus Group Discussions and Participant Observation”. According with your recommendation and literature photo-voice was established as method with the set of focus group questions, photograph activities and participant observation. (in red) Line 98-106.
Point 2: The authors use the phrase "adolescents and young people" however this is misleading as adolescents ARE young people. Recommendation is the use of the term young people as it is more inclusive of a wider age range given that the participants (i.e. parents) could be gaining insights that would be useful for their children mat any age.
Response 2: The mistake was corrected. We selected the word adolescent because represent closest the reality observed in our study. Done in the title and the whole documment.
Point 3: The presentation of the data is challenging as the reader is expected to make some assumptions about the value/importance of the quotes used.
Response 3: We included in method and results a paragraph to explain criteria for the selection of the quotes (in red) Lines 167-171 and Lines 203-367.
There is need for a little more description of each sub-section in the Results as they are not necessarily self evident as related to mental health or wellbeing. Initially both a positive and negative perspective of mental health is offered but it is unclear as to how the authors want to frame the concept. From this it would then be possible to present the rest of the data as either supports for positive mental health versus those circumstances that undermine.
Response 4: Thank you very much for your observation. We decided erased the category mental health because it will be more accurate linking with protective factors intervention and results. We will expect publish this results in another article.
Point 5: What is a Familiar Risk Factor (see 3.2.2.)? Familiar to whom? or is it menat to be Familial?.
Response 5: The mistake was corrected (in red) Line 271.
Point 6: In the conclusion the authors claim "A psychoeducational program that provides information about the relevant skills for the risk that adolescents and young people face would be beneficial." While at one level this could be read as a way to build young people's resilience. However another intervention could be about removing or reducing those factors that cause preventable negative mental health issues.
Response 6: Thank you very much your suggestion was included in the conclusions. (in red) Line 512-513.

Reviewer 3 Report
Dear Authors,
It was a great pleasure to read your article. The topic is very actual and extremely important. But I have some suggestions on how to improve the presentation of the results.
1. The Introduction and the list bibliography are too short. I have the feeling that your considerations are not properly supported by the theory. According to the topic of the article, we need information about mental health, adolescents, the role of parents in mental health prevention and the specifics of Rural Communities of Ecuador. As a consequence of the lack of theoretical framework also, the discussion is not perfect and needs improving.
2. The study was qualitative, but you did not give information on how the material was analysed. The research team considered themes and categories a priori. But how?
Also, you mentioned that:
„For the treatment and systematization of the information, ATLAS ti version 7.2 was used 153 with a license acquired by the Pontifical Catholic University of Ecuador”.
But then, in the result section, your analyses are like this:
„Some parents expressed their understanding of the term “mental health”, which was 172 interpreted as a condition that favors a sense of well-being, happiness and allows normal 173 development in adolescents and young people”
„The most relevant subcategories in the life of adolescents and young people from the 189 parental perspective was the constant load of stress due to external and internal demands 190 related to personal characteristics or educational institutions”
And you give some quotes as illustrations. I think we need some more precise information. What were the numbers of statements classified into given categories?
Photovoice – it was an intervention, not research. As you mentioned in the conclusions. Please add a note about it in the methods part.
Kind regards
Author Response
Response to Reviewer 3 Comments
Point 1: The Introduction and the list bibliography are too short. I have the feeling that your considerations are not properly supported by the theory. According to the topic of the article, we need information about mental health, adolescents, the role of parents in mental health prevention and the specifics of Rural Communities of Ecuador. As a consequence of the lack of theoretical framework also, the discussion is not perfect and needs improving.
Response 1: Thank you very much for your observation. We included papers that discuss theoretical framework and found risk modeling for mental health disorders. (in red) Line 46-76 and Line 440-442; 459-465; 489-491.
Point 2: 2. The study was qualitative, but you did not give information on how the material was analysed. The research team considered themes and categories a priori. But how?
Also, you mentioned that: „For the treatment and systematization of the information, ATLAS ti version 7.2 was used 153 with a license acquired by the Pontifical Catholic University of Ecuador”.
But then, in the result section, your analyses are like this:
„Some parents expressed their understanding of the term “mental health”, which was 172 interpreted as a condition that favors a sense of well-being, happiness and allows normal 173 development in adolescents and young people”
„The most relevant subcategories in the life of adolescents and young people from the 189 parental perspective was the constant load of stress due to external and internal demands 190 related to personal characteristics or educational institutions”
And you give some quotes as illustrations. I think we need some more precise information. What were the numbers of statements classified into given categories?
Response 2: Thank you very much for your observation. We decided erased the category mental health because it will be more accurate linking with protective factors intervention and results. We will expect publish this results in another article. (in red). We included also in method and results a paragraph to explain criteria for the selection of the quotes and more details about categories and subcategories (in red) Line 205-367.
Point 3: Photovoice – it was an intervention, not research. As you mentioned in the conclusions. Please add a note about it in the methods part.
Response 3: We included your recommendation in the section methods (in red) Line 103.

Round 2
Reviewer 3 Report
Thank you for updating the manuscript